# Crack Initiation Mechanism and Life Prediction of Ti60 Titanium Alloy Considering Stress Ratios Effect in Very High Cycle Fatigue Regime

**DOI:** 10.3390/ma15082800

**Published:** 2022-04-11

**Authors:** Ruixiang He, Haotian Peng, Fulin Liu, Muhammad Kashif Khan, Yao Chen, Chao He, Chong Wang, Qingyuan Wang, Yongjie Liu

**Affiliations:** 1Failure Mechanics and Engineering Disaster Prevention Key Laboratory of Sichuan Province, Sichuan University, Chengdu 610207, China; hege511@163.com (R.H.); 19102666105@163.com (H.P.); yaochen@scu.edu.cn (Y.C.); hechao@scu.edu.cn (C.H.); wangchongscu@163.com (C.W.); wangqy@scu.edu.cn (Q.W.); 2MOE Key Laboratory of Deep Earth Science and Engineering, College of Architecture and Environment, Sichuan University, Chengdu 610065, China; 3Institute of Future Transport and Cities, Coventry University, Coventry CV1 5FB, UK; ac1291@coventry.ac.uk

**Keywords:** very high cycle fatigue, Ti60 titanium alloy, fatigue failure mechanism, fatigue life prediction, fatigue strength prediction, stress ratio

## Abstract

Ultrasonic fatigue tests were performed on Ti60 titanium alloy up to a very high cycle fatigue (VHCF) regime at various stress ratios to investigate the characteristics. The S-N curves showed continuous declining trends with fatigue limits of 400, 144 and 130 MPa at 10^9^ cycles corresponding to stress ratios of R = −1, 0.1 and 0.3, respectively. Fatigue cracks found to be initiated from the subsurface of the specimens in the VHCF regime, especially at high stress ratios. Two modified fatigue life prediction models based on fatigue crack initiation mechanisms for Ti60 titanium alloy in the VHCF regime were developed which showed good agreement with the experimental data.

## 1. Introduction

Titanium alloys are widely used in aerospace applications as they have the advantages of high strength-to-weight ratio, excellent fatigue performance and corrosion resistance [1,2]. Aero-engine s blades and blisks are usually subjected to variable cyclic loads up to the very high cycle fatigue (VHCF) regime (>10^7^ cycles) [3]. Therefore, fatigue performances of titanium alloys in the VHCF regime are gaining increasing importance in the design and evaluation of aero-engine components. It has been found that the conventional design standard of a fixed fatigue limit at 10^7^ cycles cannot meet the actual service requirements of aero-engine components, because the materials will still fracture failure even under stresses below the fatigue limits when the service life exceeds 10^7^ cycles [4].

The basic research on VHCF of titanium alloys has been carried out for nearly 20 years, mainly concentrating on the characteristics of S-N curve, fracture morphology and crack initiation mechanism. It has been found that the competition of crack initiation failure mode between surface and interior in the VHCF region formed a “double line” or “two-step” S-N curve feature [5], as shown in Figure 1. The fatigue fracture section presents the characteristics of “fisheye”, which contains a rough area [6], also known as a “granular bright area” (GBA) [7], and there are a large number of small facets in the center [8]. There are no inclusions in the titanium alloy, and the failure behavior of VHCF is attributed to “SNDFCO” (subsurface non-defect fatigue crack origin) by Chai et al. [9], which is mainly attributed to the heterogeneity of internal microstructure [10]. The complex crack initiation mechanism of titanium alloys often leads to high dispersion of fatigue life [11,12]. In addition, there are significant differences in the fatigue strengths and crack initiation mechanisms of dual-phase titanium alloys with the effect of the stress ratio [13,14].

Moreover, many fatigue life prediction models have been employed to estimate the fatigue crack initiation and propagation life of titanium alloys, but the results were not satisfactory [15,16,17]. The main reason is that the life prediction models in the VHCF regime were developed based on the non-metallic inclusions usually found in high-strength steels [18], while titanium alloys have no inclusions. A few researchers have developed fatigue life prediction models for titanium alloys [19,20]. However, the models can not well predict the fatigue life in the VHCF regime.

Ti60 titanium alloy is a near α alloy made in China [21,22]. It has excellent creep and fatigue properties even at elevated temperatures, which makes it an ideal candidate material for advanced aero-engines [23,24,25,26]. Most previous studies for the Ti60 alloy have mainly focused on static mechanical behaviors [23,27,28,29,30]. There are few studies on fatigue of the alloy, and the investigation of VHCF performance has not been reported. However, it is extremely important to understand the VHCF behavior of the Ti60 alloy before it will be long term serviced in next-generation aero-engines. In this study, the VHCF behavior of the Ti60 alloy was explored by using an ultrasonic fatigue testing system with stress ratios of R = −1, 0.1, and 0.3. Fracture surfaces of failed specimens were characterized by microscopic techniques and the fatigue crack initiation and propagation mechanisms were investigated. Furthermore, the fatigue life prediction models were developed.

## 2. Experimental Procedure

### 2.1. Material

The material used in this study is a commercial Ti60 titanium alloy. The chemical composition of the alloy is given Table 1. The specimens for metallographic investigation were prepared with a conventional metallography process using an etchant (HF 3 mL: HNO_3_ 6 mL: H_2_O 90 mL), obtained from Chron Chemicals, Chengdu, China. The metallographic microstructure of the material obtained by SEM is shown in Figure 2. It can be seen that the Ti60 alloy has a near-α structure with the primary α phase, and the secondary α phase is distributed in β phase. The physical properties of Ti60 alloy was shown in Table 2, where E is the modulus of elasticity; *σ*_b_ is ultimate tensile strength; *σ*_0.2_ is yield strength; A is elongation after fracture; Z is the reduction of area.

### 2.2. Experiment

The VHCF tests were performed using an ultrasonic accelerated fatigue testing machine operating at a frequency of 20 kHz. As an accelerated fatigue experimental system, it takes a great contribution to reduce the experimental cycle time in VHCF investigation which improves the experimental efficiency significantly. Compared with the fatigue testing operating at a frequency of 20 Hz, it takes 139 h when the loading cycles reach 10^7^ cycles, while it takes only 8 min when operating at a frequency of 20 kHz. So, we take this mothed for the VHCF research. Moreover, the machine is developed by the authors’ group as shown in Figure 3 and the schematic diagram is shown in Figure 4. The symmetric tension-compression vibration loading was generated by the piezoelectric ceramic converter driven by the ultrasonic signal generator. Then the vibration amplitude was increased by the amplification rod and transmitted to the sample. The whole system is controlled by the computer. For symmetric loading, the ultrasonic fatigue vibration system is clamped on the static tensile machine which can provide different axial loadings. Therefore, different loading stress ratios can be offered. The cooling air generated by the spiral cold dryer is applied to cool the specimen surface to reduce the temperature rise caused by high-frequency vibration.

Ultrasonic fatigue testing machines work on the resonance principle. The specimens were designed to resonate longitudinally at 20 kHz with the ultrasonic fatigue testing system using an analytical method combined with the finite element method. The specific specimen drawings are given in Figure 5 at different stress ratios. When the loading stress ratio is greater than −1 (R > −1), the double thread at both ends is equipped to superimpose constant tensile stress.

The scanning electron microscope (JSM-6510LV, Jeol Ltd. Tokyo, Japan) was used to observe and analyze the morphology of fracture surfaces after the fatigue tests, to capture information on the fatigue crack initiation and propagation mechanisms involved in the VHCF regime.

## 3. Experimental Results

### 3.1. S-N Diagram

Figure 6 shows obtained experimental S-N (stress - number of cycles) data of the Ti60 alloy at various stress ratios of R = −1, 0.1, and 0.3, respectively, where the alternating stress *σ*_a_ is plotted as a function of the number of cycles to failure *N_f_*. The semi-solid symbols indicate the surface fatigue crack initiations, while the solid symbols present the subsurface fatigue crack initiations. The specimens which didn’t fracture up to 10^9^ cycles are shown by cross symbols. From this diagram, several tendencies can be observed. First, the Ti60 alloy has no traditional fatigue limit, and fatigue fracture will still occur after more than 10^7^ cycles. Second, when the fatigue life exceeds about 10^7^ fatigue cycles, the crack initiation site tends to transform from the surface to the subsurface of the specimen, as the semi-solid symbols change into solid symbols after 10^7^ cycles shown in Figure 6. Third, a stress ratio effect is apparent, that is, at the same fatigue life, the lower stress ratio corresponds to the higher fatigue strength.

The Basquin relationship (Equation (1)) was used to fit the S-N data in Figure 6.
(1)$$\sigma_{a}=a{\left(2N_{f}\right)}^{b}$$
where *σ*_a_ is the applied stress amplitude, *N_f_* is the number of cycles, namely the fatigue life, *a* is the coefficient of the fatigue strength, and *b* is the index of the fatigue strength. The obtained fitting results are shown in Table 3.

As shown in Figure 6, the S-N curve at each stress ratio shows continuously descent characteristics, and the slope of the S-N curve decreases with the increase of the stress ratio. The fatigue strength is the stress amplitude when cycles equal to 10^9^ cycles.

### 3.2. Fracture Surfaces

Examples of fracture surfaces for surface and subsurface crack initiations at each stress ratio are, respectively, given in Figure 7 and Figure 8, where the micrographs of the whole sections and crack initiation areas were taken. All fracture surfaces show three regional characteristics, corresponding to the crack initiation area (marked with I, rough area), crack stable propagation area (marked with II, more rough area with ridges and dims distributed along the path of crack propagation) and fast propagation area (marked with III, the relatively smooth area with ridges), respectively.

Figure 7a,b shows the example of the fatigue crack initiated at the surface at the stress ratio of R = −1, and no facets can be observed in Figure 7b at the crack initiation area. However, when the fatigue crack is initiated at the subsurface, as shown in Figure 8d, the facet can be found. At the stress ratios of R = 0.1 and 0.3, no matter the fatigue cracks initiated at the surface or subsurface, distinct facets can be found at the crack initiation areas, pointed out in Figure 7d,f and Figure 8d,f.

Under normal stress cyclic loading, facets and tearing topography surface (TTS) characteristics can be commonly found at the crack initiation areas in near α or α + β titanium alloys [12,31,32,33]. From Figure 7 and Figure 8, step facets and grain boundaries are observed which show that the facets were developed by boundary breaks of α grains. With the accumulation of cyclic loadings, the number of broken α grains increases and the micro-cracks appear among those close α_p_ grains which become the crack initiation area. Furthermore, the facet density increases with the increase of stress ratio.

## 4. The Effect of the Stress Ratio on the Production of Fatigue Failure

Crack initiation or propagation characteristics of many steels and aluminum alloys have been studied and indicated that the estimation method combined with fractography and fracture mechanics is feasible for exploring the fatigue failure process [9,15,34,35,36,37]. However, the effect of stress ratio on the VHCF failure process has rarely been investigated [19,31].

As shown in Figure 7 and Figure 8, the fracture surface can usually be divided into three areas. The size of regions I and II can be expressed as the equivalent size using the geometric parameter $\sqrt{area}$, initially proposed by Murakami [38]. The geometric parameter $\sqrt{area}$ represents the square root of the projection of the crack surface area perpendicular to the loading direction. The projected areas in all specimens were measured from fractographic morphologies in the SEM images using the procedure named Imagine J with its associative function. The relationships between the size of region I (or II) and the stress amplitude at different stress ratios are shown in Figure 9. At the stress ratio of R = −1, Bothe the sizes of regions I and II tend to decrease with the increase of the stress amplitude. At normal stress ratios of R = 0.1 and 0.3, the values of $\sqrt{area}$ are very discrete in relatively small ranges of the stress amplitude. Furthermore, the stress ratio has an obvious effect on the size of the crack initiation region, that is, in the same value of $\sqrt{area}$, the lower the stress ratio, the greater the corresponding stress amplitude.

The stress intensity factor (SIF) amplitude (ΔK) of the crack tip for the region I has been calculated using the value of $\sqrt{area}$ based on the equation as follows [16]:(2)$$\Delta K_{\left(I\;region\right)}=n\cdot \Delta \sigma \cdot \sqrt{\pi \sqrt{area}}\;$$
where *n* is a constant determined by the position of the crack, and *n* = 0.65 for the surface crack initiation, while *n* = 0.5 for subsurface crack initiation; Δσ is the stress amplitude. The calculations of $\Delta K_{I}$ for all specimens are shown in Figure 10. The values of $\Delta K_{I}$ are found to fluctuate near a constant at each stress ratio. The mean value of $\Delta K_{I}$ is obtained to be 4.36 $MPa\cdot \sqrt{m}$ at R = −1, and it decreases to be 3.96 $MPa\cdot \sqrt{m}$ at R = 0.1. Moreover, at R = 0.3, the mean value of $\Delta K_{I}$ is found to be 3.81 $MPa\cdot \sqrt{m}$ for specimens with surface crack initiation and 3.48 $MPa\cdot \sqrt{m}$ for those with subsurface crack initiation, respectively, which indicates the $\Delta K_{I}$ value of surface-initiated crack is greater than that of internal initiated crack, and this may be attributed to the crack closure effect [39]. Then, the standard error of the mean values and standard deviation of $\Delta K_{I}$ were calculated, and the results were in Table 4. A further conclusion can be drawn that the mean value of $\Delta K_{I}$ decreases with the increase in stress ratio.

## 5. Fatigue Strength Prediction

The mathematical model to predict the fatigue strength for titanium alloys is not available in the open literature to the best of our knowledge. Murakami [16] has developed an experiential formula to predict the fatigue strength limit using the size of defect or inclusion which is given in the equation below:(3)$$\sigma_{w}=\frac{M\left(HV+120\right)}{{\left(\sqrt{area}\right)}^{1/6}}{\left(\frac{1-R}{2}\right)}^{\alpha },\alpha =0.226+HV\times 10^{-4}\;$$
where *σ*_w_ is the limit of the fatigue strength, *HV* is the Vickers hardness, *M* is a constant dependent on the crack initiation position, and *M* = 1.45 for surface crack initiation, while *M* = 1.56 for interior crack initiation.

The model physically means that critical stress exists when the area size of the defect or inclusion is equal to a special value. When applied stresses are lower than the critical stress, the crack does not propagate. Therefore, the critical stress can be considered as the fatigue limit of the special area.

The threshold of the stress intensity factor Δ*K_th_* has a similar meaning to critical stress. However, titanium alloys have no defects or inclusions, in this case, the area of the region I can be used to define the defect or inclusion area, and fracture facets in this area can be considered as defects. The difference is that facets are developed during fatigue tests in titanium alloys while defects or inclusions have originally existed in other alloys. Murakami’s model aims to acquire the fatigue strength limit at 10^7^ cycles. The predicted cycles for VHCF will be larger than the experimental one when the fatigue life is larger than 10^7^, and the formation of the facets will consume the energy to make the prediction smaller. Therefore, it is more reasonable to add a modified factor ξ to Equation (3) for predicting the fatigue strength of a titanium alloy follows:(4)$$\sigma_{w}=\frac{\xi M\left(HV+120\right)}{{\left(\sqrt{area}\right)}^{1/6}}{\left(\frac{1-R}{2}\right)}^{\alpha },\alpha =0.226+Hv\times 10^{-4}\;$$
where ξ is a modified factor considering the distinction between the defect/inclusion and facet. The value of ξ takes 1.35, 0.75 and 0.65 when the stress ratio R = −1, 0.1 and 0.3, respectively, and the fitting results are shown in Figure 11. Especially, facets could be observed when the crack initiated from the interior at R = −1, so the predicted fatigue strength is only suitable for this case.

When R > 0, the errors of the predicted values (*σ*_w_ − *σ*_a_)/*σ*_a_ × 100% are between −15% and +15% as shown in Figure 11, which indicates the fitting results are feasible. When R = −1, the results are discrete, which is associated the high discrete data of titanium alloy fatigue [11], while the errors are between −15% and +15%, the conservative prediction results could be adopted when applying this mold at R = −1.

Therefore, the modified equation based on Murakami’s model is suitable for predicting the fatigue strength limit of metal materials without defects or inclusions. The model has great advantages because the form is simple with few parameters and the Vickers hardness of a material is easy to be obtained. However, this model lacks relationship with the fatigue life, and the prediction is inaccurate when the crack initiation area is small such as R = −1.

## 6. Fatigue Life Prediction of Internal Crack Initiation

Mayer et al. [40] have deduced an equation to predict the fatigue strength in the VHCF regime, which involves the relationship between the size of the inclusion and the stress amplitude shown as follows:(5)$$\sigma_{a}={\;C}^{-1/m}N_{f}^{-1/m}a_{0}^{-1/6}\;$$
where *σ_a_* is the stress amplitude, *N_f_* is the fatigue life, $\;a_{0}=\sqrt{\mathrm{area}}$ is the size of the inclusion, *C* and *m* are parameters depending on the material.

Based on Equation (5), let the area of the region I replace the size of the inclusion, one can have:(6)$$a_{0}=\Delta K^{2}n^{-2}\Delta \sigma^{-2}\pi^{-1}\;$$
where Δ*K* is the stress strength factor; *n* is a constant depending on the position of the crack, and *n* = 0.65 when the crack is initiated from the surface, while *n* = 0.5 when the crack is initiated from the interior; Δ*σ* is the applied stress amplitude, and Δ*σ* = *σ_a_* at R = −1, while Δ*σ* = *2σ_a_* when R > 0. In this paper, only interior crack initiation issues were investigated.

Equation (7) can be obtained by plugging Equation (6) into Equation (5):(7)$$N_{f}=C\sigma_{a}^{-n}\Delta K^{-n/3}m^{n/3}\Delta \sigma^{n/3}\pi^{n/6}$$

Equation (8) can be gotten by simplifying Equation (7):(8)$$N_{f}=C\sigma_{a}^{-2n/3}\Delta K^{-n/3}\;$$

Let *t* = −*n*/3, we can acquire:(9)$$N_{f}=C\sigma_{a}^{2t}\Delta K^{t}\;$$
where *C* and *t* are material parameters.

The parameters of *C* and *t* in Equation (9) are obtained by fitting the experimental fatigue data at different stress ratios as shown in Table 5. Moreover, the predicted fatigue lives can be calculated, which are shown with experimental ones in Figure 12.

The predicted fatigue lives are in agreement with the experimental ones shown in Figure 12, which indicates Equation (9) applies to the life prediction of the Titanium alloy.

Furthermore, another predicted fatigue life model based on Sun [41] (shown in Equation (10)) was used to make a comparison with the former model.
(10)$$N_{i}=\frac{1}{\alpha }{\left(\frac{\sigma_{a}}{\sigma_{Y}}\right)}^{-l}ln\frac{a_{FGA}}{a_{0}}\;$$
where *N_i_* is the fatigue life of the crack initiation, *σ_Y_* is the yield strength, *a_FGA_* is the size of the FGA, $a_{FGA}=\sqrt{area_{FGA}}$, *a*_0_ is the size of the inclusion, $\;a_{0}=\sqrt{area_{0}}$, *l* and α are the material parameters.

Some modifications have been carried out for our work based on the Sun’s model. The maximum size of the plastic zone *r_p_* in the case of plane strain can be expressed as Equation (11) [41],
(11)$$r_{p}=\frac{1}{6\pi }{\left(\frac{\Delta K}{\sigma_{Y}}\right)}^{2}\;$$
where *σ_Y_* is the yield strength.

In the Sun’s model [41], the inclusion, FGA and crack area after *i* (*i* = 1, 2,…, *n*) cycles are approximately treated as internal penny cracks in an infinite solid, and the value of *a_i_*-*a_i−1_* (*i* = 1, 2,…, *n*) is defined as “equivalent crack growth rate”, where *a_i_* is the positive square root of the crack area after *i* cycles, and *a_0_* is the positive square root of the inclusion projection area perpendicular to the applied stress axis. Furthermore, it is assumed that the equivalent crack growth rate in the FGA region is related to the maximum size of the plastic zone at the crack tip.

In this paper, the sizes of α_p_ and region I are used to replace the size of inclusion and FGA, respectively [38].
(12)$$a_{i}-a_{i-1}=\frac{\beta }{6\pi }\frac{\sigma_{a}^{2}\pi a_{i-1}}{\sigma_{Y}^{2}}=\frac{\beta }{6}{\left(\frac{\sigma_{a}}{\sigma_{Y}}\right)}^{2}a_{i-1},i=1,2,\ldots ,n\;$$

The equivalent crack length after *n* cycles can be given by:(13)$$a_{n}={\left(1+\frac{\beta \sigma_{a}^{2}}{6\sigma_{Y}^{2}}\right)}^{n}a_{0}\;$$

From Equation (12), the fatigue life *N*_I_ consumed in the region I can be obtained by the minimum of *n*.
(14)$$a_{n}\geq a_{I}\to {\left(1+\frac{\beta \sigma_{a}^{2}}{6\sigma_{Y}^{2}}\right)}^{n}\geq \frac{a_{I}}{a_{0}}\;$$

The Equation (14) can be approximately solved as,
(15)$$N_{i}=\frac{1}{ln\left(1+\beta^{\prime }\right)}ln\frac{a_{I}}{a_{0}}\;$$
where *β^′^* = $\frac{\beta \sigma_{a}^{2}}{6\sigma_{Y}^{2}}$.

Since the fatigue life consumed in crack initiation is very long in the VHCF regime, the value of *β^′^* is much smaller than one. Taking ln (1 + *β^′^*) in Equation (15) for the Taylor series expansion, we can have,
(16)$$N_{i}=\frac{1}{\beta^{\prime }}ln\frac{a_{I}}{a_{0}}\;$$

According to the analysis of Sun [39], we can satisfy the Equation (16) and a new fatigue life prediction model for the Ti60 alloy in the VHCF regime can be derived as,
(17)$$N_{i}=\frac{1}{\alpha }{\left(\frac{\sigma_{a}}{\sigma_{Y}}\right)}^{-l}ln\frac{a_{I}}{a_{0}}$$

The values of *α* and *l* at different stress ratios can be obtained by fitting the experimental data and the results are given in Table 6.

The predicted fatigue lives are compared with the experimental ones shown in Figure 13. The data disperse around the line *y* = *x*, which means the predicted fatigue live are well in agreement with the experimental ones, while the dispersion of the former model is relatively greater. So, the fatigue life prediction for the Ti60 alloy can also be carried out by Sun’s model.

From the analysis of the two models above, we can conclude that the size of the region I which has several facets has a similar physical meaning as the size of defects or inclusions, so the region I area can be used to replace the defect or inclusion area for titanium alloys in VHCF studies.

## 7. Conclusions and Perspectives

### 7.1. Conclusions

Ultrasonic fatigue tests were performed on the Ti60 titanium alloy in a very high cycle fatigue regime at different stress ratios, main conclusions are drawn as follows:(1)The S-N curves with different stress ratios show continuous declining tendencies. Fatigue cracks found to be initiated from subsurface of the specimens, and the tendency of internal crack initiation increases when R > 0.(2)Based on SEM observations on fracture surfaces, the whole progress of fatigue failure can be divided into four regions: (I) crack initiating stage involving a large number of small flat rough areas; (II) highly rough areas containing radial ridges; (III) a relatively flat and wide area containing radial stripes; and (IV) an area containing obvious dimples.(3)The Δ*K*_I_ of internal crack initiation is much larger than that of surface crack initiation. The mean values of Δ*K*_I_ are 4.36, 3.57 and 3.49 MPa$\sqrt{m}$ at stress ratios of −1, 0.1 and 0.3, respectively. The difference in the Δ*K*_I_ can be explained by the crack closure effect.(4)Two modified fatigue strength prediction models on the Ti60 titanium alloy are developed and in good agreement with experimental results.

### 7.2. Perspectives

Based on the existing research, the following aspects can be further explored:(1)The very high cycle fatigue regime for Ti60 titanium alloy at high temperature especially at 600 °C need to be researched for its the service environment of Ti60 titanium.(2)The temperature with stress ratio effect on Ti60 titanium alloy at a very high cycle regime also need to be researched, the comprehensive influence factor for a very high cycle regime should be further researched.

## Figures and Tables

**Figure 1 materials-15-02800-f001:**
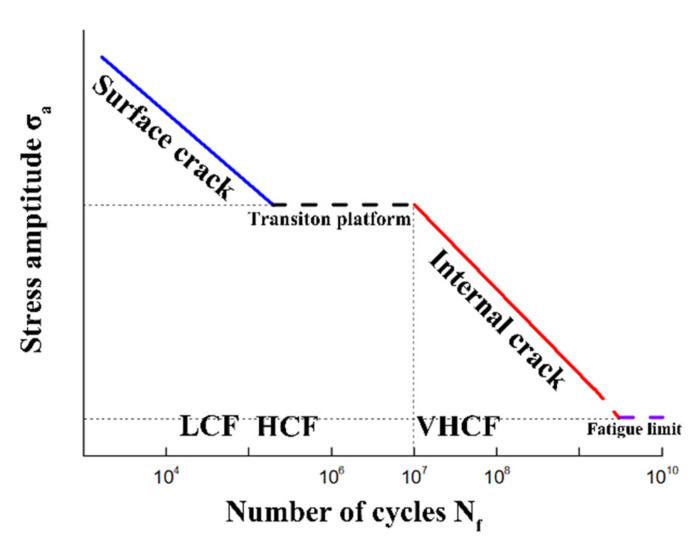
The schematic diagram of two step S-N (stress-number of cycles) curve. LCF, HCF, VHCF indicate low, high, very high cycle fatigue, respectively.

**Figure 2 materials-15-02800-f002:**
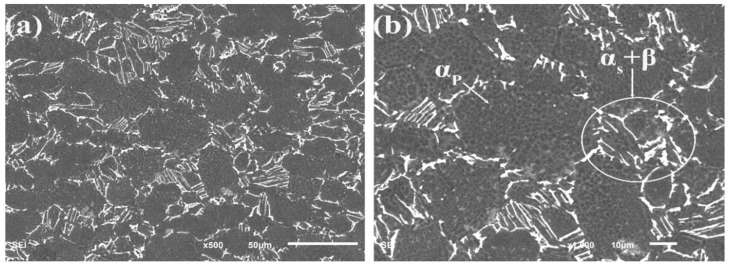
(**a**) Metallographic microstructure; (**b**) the high-magnification photograph of microstructure, α_s_ + β indicates the secondary α phase distributed in β phase.

**Figure 3 materials-15-02800-f003:**
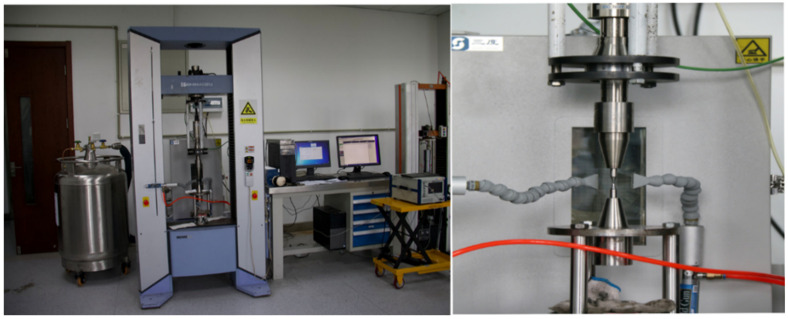
Ultrasonic fatigue testing machine [31].

**Figure 4 materials-15-02800-f004:**
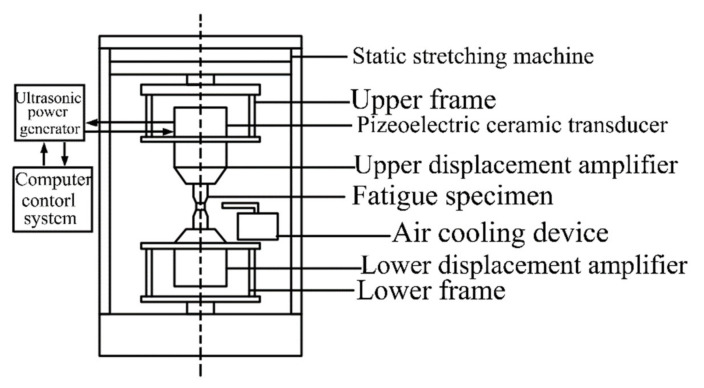
The schematic diagram of asymmetric ultrasonic fatigue.

**Figure 5 materials-15-02800-f005:**
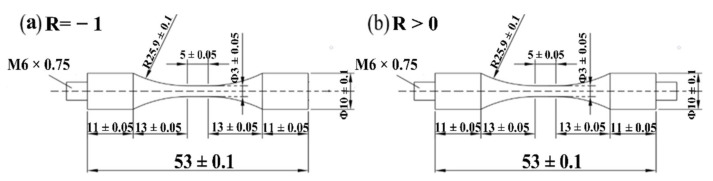
The dimensions of VHCF specimen: (**a**) R = −1; (**b**) R > 0, (unit: mm).

**Figure 6 materials-15-02800-f006:**
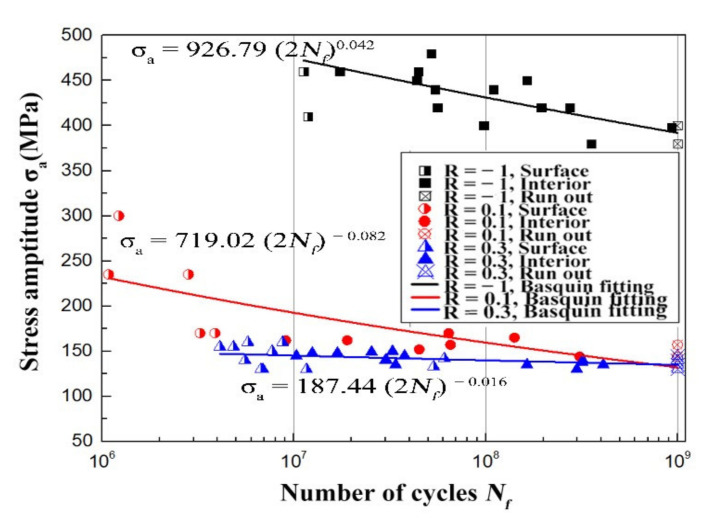
S-N diagram at R = −1, 0.1 and 0.3.

**Figure 7 materials-15-02800-f007:**
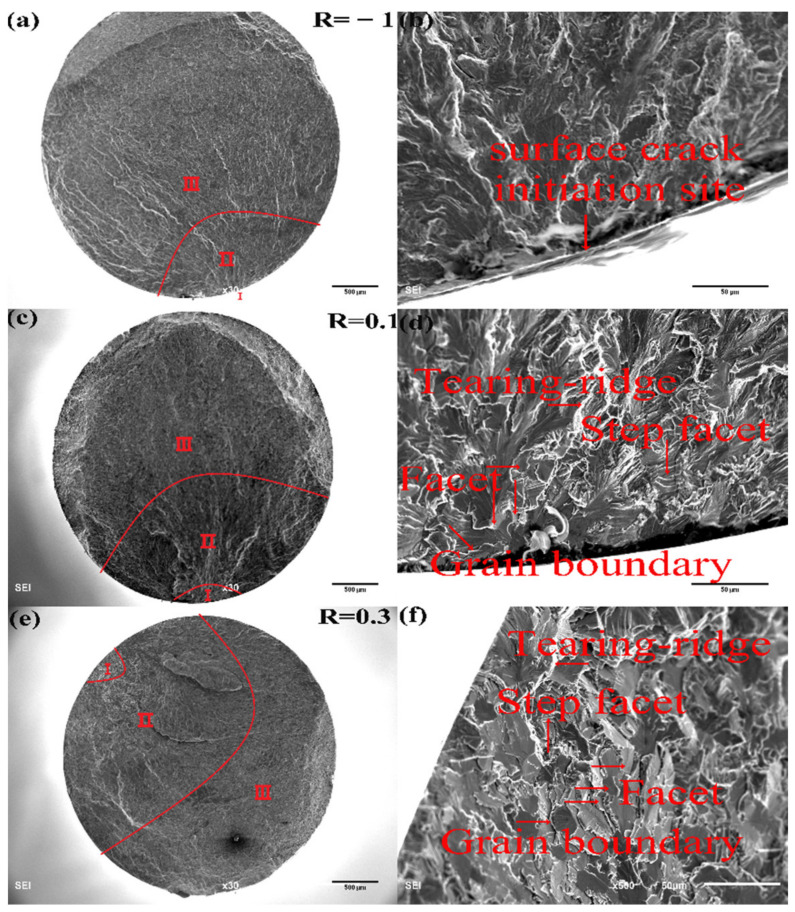
Fracture surfaces of specimens with surface crack initiation: (**a**,**b**) R = −1, *σ*_a_ = 460 MPa, *N_f_* = 1.77 × 10^7^ cycles; (**c**,**d**) R = 0.1, *σ*_a_ = 300 MPa, *N_f_* = 1.224 × 10^6^ cycles; (**e**,**f**) R = 0.3 *σ*_a_ = 148 MPa, *N_f_* = 1.687 × 10^7^ cycles. I, II and III indicate the region I, II and III, respectively.

**Figure 8 materials-15-02800-f008:**
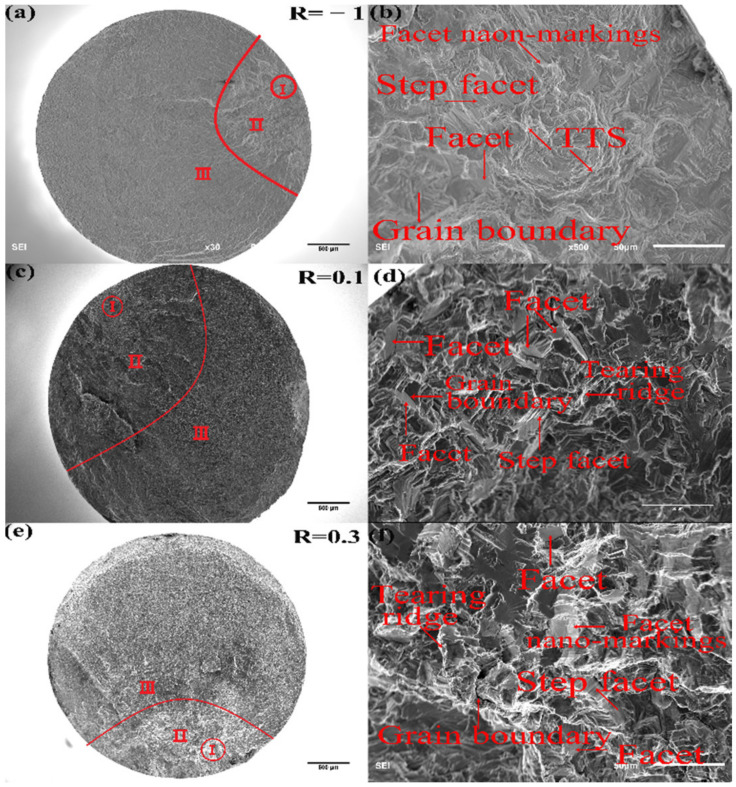
Fracture surface of specimens with subsurface crack initiation: (**a**,**b**) R = −1, *σ*_a_ = 460 MPa, *N_f_* = 1.212 × 10^7^ cycles; (**c**,**d**) R = 0.1, *σ*_a_ = 170 MPa, *N_f_* = 6.402 × 10^7^ cycles; (**e**,**f**) R = 0.3, *σ*_a_ = 130 MPa, *N_f_* = 2.987 × 10^8^ cycle. I, II and III indicate the region I, II and III, respectively.

**Figure 9 materials-15-02800-f009:**
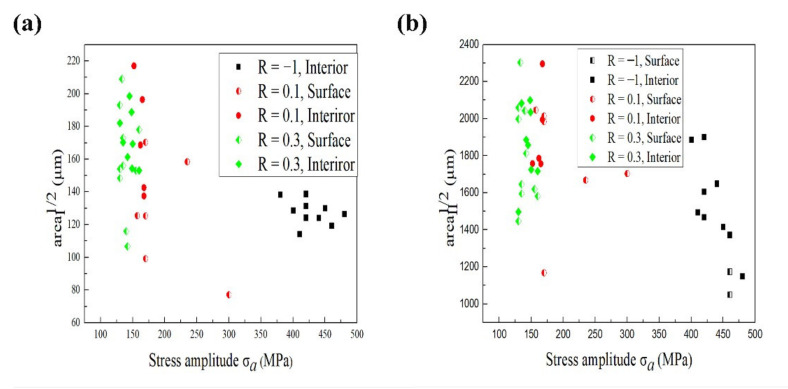
Relationships between value of $\sqrt{area}$ and stress amplitude: (**a**) region I; (**b**) region II.

**Figure 10 materials-15-02800-f010:**
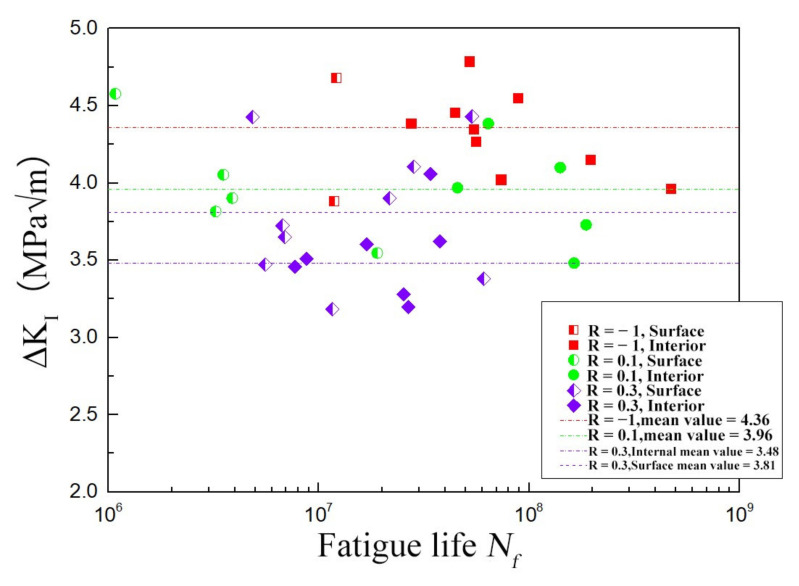
Values of $\Delta K_{I}$ at different conditions.

**Figure 11 materials-15-02800-f011:**
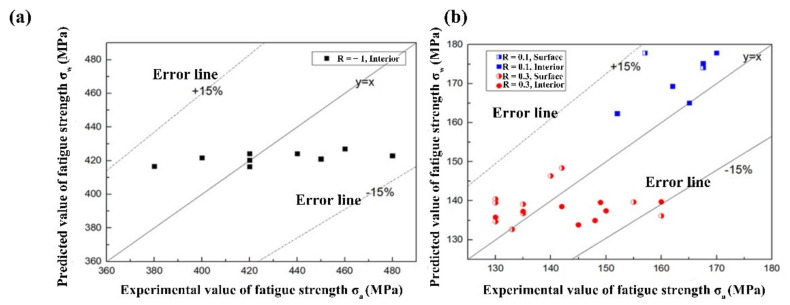
Comparison between the predicted value and experimental value of fatigue strength (**a**) R = −1, (**b**) R > 0.

**Figure 12 materials-15-02800-f012:**
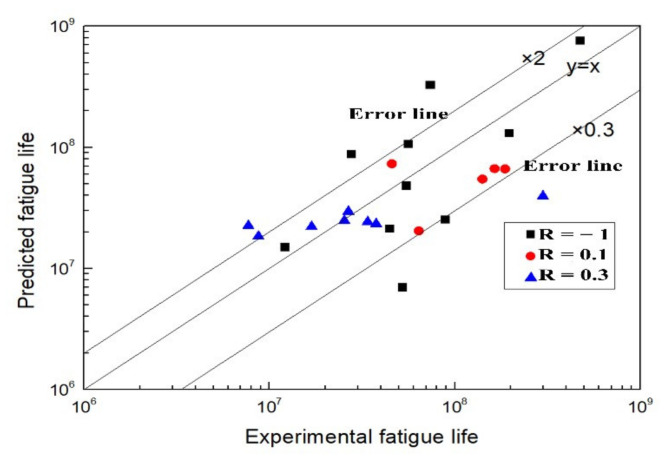
Comparison between predicted and experimental fatigue lives.

**Figure 13 materials-15-02800-f013:**
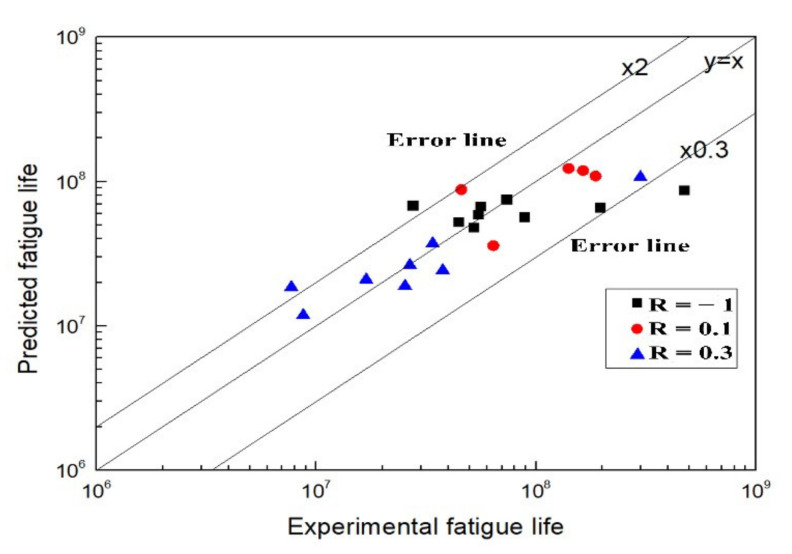
Comparison between predicted and experimental fatigue lives.

**Table 1 materials-15-02800-t001:** Chemical composition (wt. %).

Material	Ti	Al	Sn	Zr	Mo	Si	Ta	C
Ti60	84.84	5.6	4.0	3.5	1.0	0.5	0.5	0.06

**Table 2 materials-15-02800-t002:** The physical properties of Ti60 alloy.

Alloy	E/GPa	*σ*_b_/Mpa	*σ*_0.2_/Mpa	A/%	Z/%
**Ti60**	114	1044	934	11	23

**Table 3 materials-15-02800-t003:** Results of *S-N* curves fitting by the Basquin3 relationship.

Stress Ratio	R = −1	R = 0.1	R = 0.3
*a*	926.79	719.02	187.44
*b*	−0.042	−0.082	−0.016
Fatigue limit (MPa)	380	144	125

**Table 4 materials-15-02800-t004:** Results of $\Delta K_{I}$ ($MPa\cdot \sqrt{m}$).

Stress Ratio	R = −1	R = 0.1	R = 0.3 (Surface)	R = 0.3 (Interior)
Mean value	4.36	3.96	3.81	3.48
standard error	0.021	0.034	0.049	0.041
Standard deviation	0.073	0.107	0.148	0.136

**Table 5 materials-15-02800-t005:** The parameters of *C* and *t* at different *R*.

*R*	*C*	*t*
−1	1.11565 × 10^50^	−7.14963
0.1	1.52153 × 10^15^	−1.47486
0.3	2.21737 × 10^14^	−1.42973

**Table 6 materials-15-02800-t006:** The values of *α* and *l* at different stress ratios.

*R*	*α*	*l*
−1	4.11 × 10^−7^	2.44709
0.1	1.32 × 10^−11^	−4.36986
0.3	0.0488	6.6042

## Data Availability

All the data is available within the manuscript.

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
