# Peer review of "Crack Initiation Mechanism and Life Prediction of Ti60 Titanium Alloy Considering Stress Ratios Effect in Very High Cycle Fatigue Regime"

_materials, 2022, doi:10.3390/ma15082800_

Round 1

Reviewer 1 Report

This manuscript sets out an experimental study on fatigue life of Titanium alloy Ti60. The method of using ultrasonic fatigue testing is interesting and is useful for gathering data in the ultra high fatigue test regime. 

Raw data was gathered on the fatigue life of the specimens. The fatigue limit was taken at 10^9 cycles. After fracture, surfaces were investigated for their fracture surface characteristic. Fitting exercises were undertaken to determine the fatigue prediction of the material.

However, the document requires significant English editing and proof reading before it can be considered for publication. 

I would also make the following comments:

  1. Please rewrite the introduction in the present tense (or present perfect tense) when discussing literature of others. Perhaps a diagram explaining a 'double-line' or 'two-step' S-N curve would be helpful.
  2. What are the composition requirements of Ti60 and does the material tested here lie within the compositional requirement? 
  3. In the introduction, the authors state: "the conventional design standard of a fixed fatigue limit at 10^7 cycles cannot meet the actual service requirements of aero-engine components..." Please explain this further. does this mean that 10^7 is not a true fatigue limit. Does the material even exhibit a fatigue limit?
  4. SNDFCO is written in an unusual font size. why?
  5. In the main text of the abstract, write out minus 1 for the stress ratio. the term -1 is taken as a hyphen and doesn't read well in the fist instance. The minus symbol is on a line the number 1 is on the next line.
  6. Why didn't the authors attempt to generate a Goodman-Haigh diagram to determine if the fatigue prediction follows a  Goodman, Soderberg, or Gerber line?
  7. Check the caption on table 2. should start with a capital T.
  8. What microscopic method was used capture figure 1? Optical or electron? BF or DF, or SE or BSE. The image does not look to be of  publication standard. The scale bar is difficult to read. There is no annotation to point out the different features and phases. The description given in the main text and the caption are inadequate. In the main text it says "has a near-α structure with the primary α phase, and the secondary α phase is distributed in β phase." This is very confusing. Perhaps a Brightfield optical image alongside the current image would be helpful.
  9. The Basquin relationship is used to define the S-N curves. It would be worthwhile noting that 2Nf is the number of reversals. Nf is called up as number of cycles in figure 4 but is called fatigue life in equation 1. Please be consistent in naming the variables.
  10. Is there really a fatigue limit with this material? For example has the S-N curve leveled off after 10^9 cycles. this doesn't appear to be the case for R= -1.
  11. How accurate is the cycle counting methods. I assume that at such high frequency it is difficult to count the numbers of cycles accurately. Please explain how this was done and any error associated with the approach.
  12. Figures 5 and 6 are difficult to read. The contrast is such that it is difficult to read the annotations, try using a different colour such as red for the annotations. sometimes scale bars are on the image (typical) and sometimes below the image (untypical). Please be consistent in the placing of scale bars (make them clearer). 
  13. The discussion on fracture surface might be improved if the surface are described in the context of intergranular versus transgranular fracture.
  14. DeltaK I values are given in figure 9 . Can you also give Standard error of the mean values and standard deviation? Again be consistent with naming Nf
  15. Write Vickers hardness as HV.
  16. Does figure 10 (a) really show a trend? 
  17. It was stated that "From the analysis for the two models above, we can conclude that the size of region â…  which have lots of facets has the similar physical meaning as the size of defects or inclusions, so the region â…  area can be used to replace the defect or inclusion area". Please justify this statement.

Reviewer 2 Report

The article highlights peculiarities of ultrasonic fatigue tests performed on Ti60 titanium alloy up to a very high cycle fatigue (VHCF) regime at various stress ratios. The fatigue behaviors at stress ratios of -1, 0.1 and 0.3, respectively had been investigated. Two modified fatigue life prediction models based on fatigue initiation mechanisms were developed.

The article is interesting, but a number of shortcomings need to be corrected:

  1. The text in Fig.1 cannot be recognized.
  2. The scale bars in Fig.5 and Fig.6 cannot be recognized.
  3. The font size in Fig.3, Fig.8, and Fig.10 should be increased.
  4. It is unclear in Fig.5a, b where is the place of crack nucleation on the sample surface. It seems that zone I is the light spot located outside the sample.
  5. According to the authors, there are very few papers on evaluation methods using fractographic studies and fracture mechanics approaches using different load cycle asymmetries. Therefore, references to such works should be added, in particular https://doi.org/10.1007/s11003-009-9177-4.
  6. For a more effective analysis of the data presented in Fig.8, it is suggested that legends (color and symbols) in Fig.8a and Fig.8b be the same.
  7. The numbering of formulas should be corrected because of duplication of the number (1) on page 4 and page 7.
  8. Since in fracture mechanics KI means the stress intensity factor in mode I, it is suggested to use K (I region) in the formula (1) on page 7.
  9. The value of ξ is given for R=0.75 on page 8: “Where ξ is a modified factor considering the distinction between the defect/inclusion and facet. The value of ξ takes 1.35, 0.75 and 0.65 when the stress ratio R=-1, 0.75 and 0.1 respectively, and the fitting results are shown in Fig. 10.”. However, the value R=0.75 was not considered before. What it is given for? Instead of this, the value of ξ is required for R=0.3.
  10. Predicting fatigue lives according to the formula (8) (Fig. 11) for R=0.3 is not correct. It should be noted in the text of the manuscript that it is impractical to use the formula (8) to determine fatigue lives at R=0.3 (Fig. 11).

Reviewer 3 Report

Title: Crack initiation mechanism and life prediction of Ti60 titanium alloy…

Manuscript ID: materials-1651535

Authors: He et al.

Dear Authors,

Thank you for the opportunity to read your article. I found the topic is interesting. Generally speaking, there are some results presented in order to capture some trends. On the other hand, the methods and results need more clear explanation and discussion. I suggest this article be revised before resubmission for another review process if any. As a conclusion, I recommend its major revision at this state.

I hope my comments are helpful.

Good luck,

A reviewer

Major concerns:

“Abstract”

-It seems that the abstract is too short and lack of important information. Please consider adding more information suggested by the journal.

https://www.mdpi.com/journal/materials/instructions

“Keywords”

-Please consider listing keywords that are not used in the article title.

“1. Introduction”

-“Titanium alloys…high strength-to-weight ratio…corrosion resistance.”->Please consider citing some numbers and references to support your statement.

-“…granular bright area (GBF)…”-> …granular bright area (GBA)…?

-In introduction or materials and methods, please clearly state why you selected this specific method and conditions “…using an ultrasonic fatigue…0.3.”.

“2. Experimental procedure”

“2.1. Material”

-Table 1: “Ti balance”->Do you mean 100 - sum of other elemental%? If yes, please consider stating a certain value other than balance.

-Table 2: Please define those physical properties, i.e. E, σb, σ0.2, A, Z before using their symbols.

-Tables 1 and 2 (and elsewhere)->Please describe and discuss the results.

-Figure 1: “…Ti60 alloy has a near-α structure…the secondary α phase is distributed in β phase.”->In Figure 1, please consider pointing out which points represent α phase or β phase.

“2.2. Experiment”

-Please name relevant references of your equipment in case a reader wishes to know more about it.

-“…a frequency of 20 kHz.”->Please consider providing your justification of this specific frequency and stating what is the potential effect of the frequency on the fatigue behavior of your specimens.

-Figure 2: In the right figure, please consider pointing out and explaining responsibility of each part of your equipment. For example, which part of the equipment generate “tension-compression vibration load”?

-Figure 3: It seems that the both (a) and (b) are exactly same. Please check and revise them if necessary.

-“…SEM was used…”->-In this section, please consider providing more details about the measurements. For example, please consider providing more detail information about your SEM imaging, including the way you deposited your sample(s) in an SEM chamber, detector type (SE? BSE?), accelerating voltage, working distance. Also, please consider explaining how you analyzed your images (e.g. area measurement). Those information would be helpful for future researchers. This comment also applied to all the characterization methods introduced in this section.

doi.org/10.1016/j.actamat.2005.12.014

doi:10.3390/electronics8101202

“3. Experimental results”

-Generally speaking, if you show a figure/table, please cite, explain and discuss it. Otherwise, a reader will never understand your message (unless your figure/table is totally self-explanatory).

-Also, please make sure to link your statements and your results. You can find more detail comments below.

“3.1. S-N diagram”

-Figure 4: There are 3 data sets. Please consider explaining and discussing all of them.

-“…the crack initiation site tends to transform from the surface to the subsurface of the specimen.”->Please consider stating how your data tell this information.

“3.2. Fracture surfaces”

-Figures 5 and 6: (1) In the section 3.2, please clearly inform what are the difference between those figures. (2) Please describe and discuss all the results according to the experimental conditions.

“4. The effect of the stress ratio on the production of fatigue failure”

-Figure 8: “area1/2” was plotted. In the materials and methods section, please consider explaining how you measured the area of corresponding regions (I, II, III).

-“…flucturate…”->…fluctuate…

“5. Fatigue strength prediction”

-Equation (2) (and elsewhere): Please define all the symbols used in this equation.

-“…titanium alloys have no defects or inclusions…the area of region I can be used to define the defect or inclusion area, and fracture facets in this area can be considered as defects.”->Please consider explaining more about this point and provide your justification of your assumption.

-“…Fig.10, which indicates the fitting results are feasible.”->Please consider revising this statement. Especially for Figure 10(a), there is no good correlation between your prediction and experimental values of the fatigue strength.

“6. Fatigue life prediction of internal crack initiation”-> 6. Fatigue life prediction incorporating internal crack initiation?

-“…the size of region I which have lots of facets…the size of defects or inclusions…”->Please consider citing your results and figures/tables to support your statement.

“7. Conclusions”

-You may state some future perspectives.

Minor concerns:

English needs to be further polished. You may use some of my comments above for this purpose.

Round 2

Reviewer 1 Report

I have read through the response and the new manuscript. Generally it is improved, but there are still a few issues before it can be recommended for publication.

Figure 1 is copied from another journal. This image will be protected by copyright and should not be published in MDPI unless you have permission from that source of information. Generally speaking it is not acceptable to copy someone else's diagram and use it in your own publication, unless you are writing up a review paper. 

I am still unhappy with figure 2. Is this two figures or one. I cannot tell.  The error bars are almost indistinct. Use borders if you want to show show two areas. Use a background area around the error bar to make it stand out. There is black text on the diagram but I can hardly read it. I strongly recommend that you  look at other highly cited materials papers and look at how they represented their micrographs. Choose a good style and try to follow it.

I still don't believe that figure 11 (a) is following a trend even though it is within the +/-15% error. Perhaps if you plotted all of the data in figure 11 (a) and (b) on one graph, the result might seem clearer.

Reviewer 2 Report

The authors took into account almost all comments of the reviewer and made appropriate corrections to the manuscript. However, a number of shortcomings need to be corrected:

The font size in Fig.5 should be increased.

The font size in legend in Fig.6, Fig.10, and Fig.11 should be increased.

Reviewer 3 Report

Dear Authors,

As all the comments and concerns were addressed, I suggest that the journal accepts the manuscript for its publication.

Best regards,

A reviewer

Author Response

Special thanks to you for your good comments again.